

**Impact of low-pressure systems on winter heavy air pollution in the northwest Sichuan Basin, China**
Guicai Ning[1], Shigong Wang[2,1], Steve Hung Lam Yim[3,4], Jixiang Li[1], Yuling Hu[1], Ziwei Shang[1], Jinyan Wang[1], Jiaxin
Wang[2]
[1]The Gansu Key Laboratory of Arid Climate Change and Reducing Disaster, College of Atmospheric Sciences,
Lanzhou University, Lanzhou 730000, China
[2]Key Laboratory of Education Bureau of Sichuan Province for Mountain Environmental Meteorology and Public
Health, School of Atmospheric Sciences, Chengdu University of Information Technology, Chengdu 610225, China
[3]Department of Geography and Resource Management, The Chinese University of Hong Kong, Hong Kong, China
[4]The Institute of Environment, Energy and Sustainability, The Chinese University of Hong Kong, Hong Kong, China
Correspondence to: Shigong Wang (wangsg@cuit.edu.cn, wangsg@lzu.edu.cn)

**Abstract**

The cities of Chengdu, Deyang, and Mianyang in the northwest Sichuan Basin are part of a rapidly developing
urban agglomeration adjoining the eastern slopes of the Tibetan Plateau. Heavy air pollution events have frequently
occurred over the cities in recent decade, but the effects of meteorological conditions on these pollution events are
unclear. We explored the effects of weather systems on winter heavy air pollution from 1 January 2006 to 31 December
2012 and from 1 January 2014 to 28 February 2017. Ten heavy air pollution events occurred during the research period
and eight of these took place while the region was affected by a dry low-pressure system at 700 hPa. When the urban
agglomeration was in front of the low-pressure system and the weather conditions were controlled by a warm southerly
air flow, and a strong temperature inversion appeared above the atmospheric boundary layer acting as a lid. Forced by
this strong inversion layer, the local secondary circulation was confined within the atmospheric boundary layer and the
horizontal wind speed in the lower troposphere was low. As a result, vertical mixing and horizontal dispersion in the
atmosphere were poor, favoring the formation of heavy air pollution events. After the low-pressure system had
transited over the region, the weather conditions in the urban agglomeration were controlled by a dry and cold air flow
from the northwest at 700 hPa. The strong inversion layer gradually dissipated, the secondary circulation enhanced and
uplifted, and the horizontal wind speed in the lower troposphere also increased, resulting in a sharp decrease in the



concentration of air pollutants. The strong inversion layer above the atmospheric boundary layer induced by the
low-pressure system at 700 hPa thus played a key role in the formation of heavy air pollution during the winter months
in this urban agglomeration. This study provides scientific insights for forecasting heavy air pollution in this region of
China.
**1 Introduction**
Air quality, especially the occurrence of heavy air pollution events, is not only strongly affected by excessive
emission of air pollutants, but is also closely associated with meteorological conditions, including atmospheric
circulations, weather systems, structures of atmospheric boundary layer, and the corresponding meteorological
parameters (Ye et al., 2016;Zhang et al., 2012a;Wei et al., 2011;Deng et al., 2014;Li et al., 2015;Gu and Yim, 2016).
The total amount of pollutants emitted in a particular period of time is usually stable in China (Wu et al., 2017), but
there are large differences in the concentrations of air pollutants, indicating that the meteorological conditions have an
important role in modulating concentrations of ambient air pollutants (Wang et al., 2009;Wang et al., 2010;Yang et al.,
2011;Gao et al., 2011;Ji et al., 2012;Hu et al., 2014;Ji et al., 2014).
Weather systems control the ability of the atmosphere to disperse pollutants and thus provide the primary driving
force for variations in regional air pollution (Ye et al., 2016;Chen et al., 2008). Leśniok et al. (2010) reported that the
atmosphere was stagnant and that the concentrations of near-ground air pollutants increased significantly in Upper
Silesia, Poland during periods with an anticyclonic circulation. By contrast, when a cyclonic circulation prevailed,
causing an inflow of fresh air masses from regions with lower levels of pollution, the concentrations of air pollutants
decreased. As synoptic-scale high-pressure ridges at 500 hPa transit across Utah, accompanied by warm advection
above valleys, the stability of the atmosphere is increased and favors the formation of persistent pools of cold air,
resulting in deterioration in air quality (Whiteman et al., 2014).
Many studies have been carried out on the impact of weather systems on air quality in China. Bei et al. (2016)
classified typical synoptic situations and evaluated their contributions to air quality in the Guanzhong Basin, China.



They found that an inland high-pressure system at 850 hPa resulted in temperature inversion, low horizontal wind
speed and a shallow atmospheric boundary layer, which favor the formation of heavy air pollution. Weather systems
have significantly impact on the transport of air pollutants. Luo et al. (2018) reported that the trans-boundary air
pollution and the pollutant concentration in Hong Kong increased when a tropical cyclones is approaching. During
winter, floating dust particles over northwestern China can be carried downstream to northern China by the prevailing
northwesterly winds at 700 hPa, where they mix with anthropogenic pollution to form a regional haze (Tao et al.,
2014;Tao et al., 2012). Changes in weather systems also significantly influence air quality. Shallowing of the East
Asian trough and weakening of the Siberian high-pressure in winter can induce weak horizontal advection and vertical
convection in the lower troposphere, reducing the height of the boundary layer in the Beijing–Tianjin–Hebei region
and favoring the formation of haze (Zhang et al., 2016).
The deep Sichuan Basin to the east of the Tibetan Plateau has a maximum elevation difference >2000 m, and is
ranked fourth in China for heavy air pollution after the Beijing–Tianjin–Hebei region, the Yangtze River Delta, and the
Pearl River Delta (Zhang et al., 2012b;Tian et al., 2017). The complex terrain leads to unique weather systems that
affect air quality in this region (Chen et al., 2014;Huang et al., 2017). Low-pressure systems, such as a southwest
vortex and low trough, are often formed at 700 hPa due to the dynamic and thermodynamic effects of the Tibetan
Plateau (Wang and Tan, 2014;Yu et al., 2016) and have different characteristics in different seasons. They are warm
and moist low-pressure systems in summer and autumn and have crucial effects on local precipitation (Peng and
Cheng, 1992;Feng et al., 2016); much work has been carried out in an attempt to understand the impacts of these
low-pressure systems on precipitation (Kuo et al., 1988;Kuo et al., 1986;Chen et al., 2015;Ni et al., 2017;Fu et al.,
2011). In winter and spring, however, these low-pressure systems are both dry and cold (Feng et al., 2016). No attempt
has previously been made to investigate the association between air quality and these dry and cold low-pressure
systems.
Chengdu, Deyang, and Mianyang, have undergone rapid development to form an urban agglomeration in the



northwest Sichuan Basin. This urban agglomeration lies close to the eastern slopes of Tibetan Plateau, and is affected
by low-pressure systems moving east from the plateau (Feng et al., 2016). Heavy air pollution events have frequently
occurred over there in recent decade. Number of days with exceedance of Grade II standards (MEP, 2012) is more than
150 days each year in Chengdu (Ning et al., 2018). Most previous studies have investigated the basic characteristics of
air pollution (Luo et al., 2001;Chen and Xie, 2012;Tao et al., 2013a;Tao et al., 2013b;Chen et al., 2014;Zhang et al.,
2017;Ning et al., 2018) and the related meteorological parameters (Li et al., 2015;He et al., 2017;Liao et al., 2017;Zeng
and Zhang, 2017). However, the influencing mechanism of dry low-pressure system on heavy air pollution events has
yet to be comprehensively explored. The main purpose of this study was to statistically analyze the relationships
between low-pressure systems and winter heavy air pollution events in this urban agglomeration, and to explore the
physical mechanisms involved in the formation of winter heavy air pollution. This study can deepen our understanding
of the meteorological causes of heavy air pollution events in winter, and provide scientific insights that can be used by
local governments to take effective measures to mitigate air pollution.

This paper is organized as follows. The data and methods are described in Section 2. Section 3 provides a

statistical analysis of the relationships between the low-pressure systems and winter heavy air pollution. Section 4
illustrates the physical mechanisms of the effect of weather systems on air pollution and our conclusions are
summarized in Section 5.
**2 Data and methods**
**2.1 Air quality data**

Air pollution in the Sichuan Basin during the winter months is mainly caused by particulate matter (Ning et al.,

2018). The Chinese Ministry of Environmental Protection (MEP) currently monitors particles with diameters $\leq 2.5$ μm
($PM_{2.5}$) and particles with diameters $\leq 10$ μm ($PM_{10}$). We studied heavy air pollution events occurring during the winter
months in Chengdu, Deyang, and Mianyang in the northwest Sichuan Basin (**Fig. 1**). We selected pollution events with
a daily $PM_{10}$ mean concentration $\geq 350$ μg m$^{-3}$ from 1 January 2006 to 31 December 2012 and from 1 January 2014 to



28 February 2017. The third revision of the "Ambient Air Quality Standard" (AAQS) (GB3095-12) was released on
February 29$^{th}$, 2012, and $PM_{2.5}$ was adopted into the AAQS in China since 2013. The air quality monitoring stations
needed to be updated and the data of air pollutants monitored in the three cities existed missing measurement during
2013. Thus, the winter heavy pollution events during 2013 had not been analyzed in this paper. The $PM_{10}$ daily mean
concentration from 1 January 2006 to 31 December 2012 refers to the 24-hour average concentration of $PM_{10}$ from
12:00 BST (Beijing Standard Time, i.e., Coordinate Universal Time (UTC) +8 h) on the previous day to 12:00 BST on
the current day. The $PM_{10}$ daily mean concentration from 1 January 2014 to 28 February 2017 refers to the 24-hour
average concentration of $PM_{10}$ from 00:00 BST to 24:00 BST on the current day. Hourly concentrations of $PM_{2.5}$,
sulfur dioxide ($SO_2$), nitrogen dioxide ($NO_2$), carbon monoxide (CO), and ozone ($O_3$) were also measured in the three
cities from 1 January 2014 to 28 February 2017. These above air quality data were collected from the MEP website
(http://datacenter.mep.gov.cn/index).
**2.2 Meteorological data**
**(1) ERA-Interim daily data**
To analyze the weather systems at 700 hPa, and the dynamic and thermodynamic conditions in the lower
troposphere, the temperature, the geopotential, the vertical velocity, and the u and v components of wind during the
study period were obtained from the ERA-Interim daily dataset (0.125 °× 0.125 °grids) from 950 to 500 hPa for a total
of 14 vertical layers (with a vertical separation of 25 hPa from 950 to 775 hPa and a vertical separation of 50 hPa from
750 to 500 hPa). These meteorological data are available for 00:00, 06:00, 12:00, and 18:00 UTC and were collected
from the website (http://apps.ecmwf.int/datasets/data/interim-full-daily/levtype=pl/). The height of the atmospheric
boundary layer was obtained from the ERA-Interim daily dataset at the surface with a 3 h temporal resolution (00:00,
03:00,      06:00,      09:00,      12:00,      15:00,      18:00,      and      21:00      UTC)
(http://apps.ecmwf.int/datasets/data/interim-full-daily/levtype=sfc/) to explore the structure of the atmospheric
boundary layer.





**(2) Sounding data**

Radiosonde measurements from launches at Wenjiang station (see **Fig. 1**) in Chengdu city (30.70 °N, 103.83 °E, elevation 541.0 m) at 08:00 and 20:00 BST were obtained from the University of Wyoming website (http://weather.uwyo.edu/upperair/sounding.html) and included the temperature, potential temperature, and horizontal wind. These data were used to investigate the dynamic and thermodynamic structure of the lower troposphere.

**(3) Visibility**

Visibility from three observation stations in the three cities was provided by the National Meteorological Information Center of the China Meteorological Administration, and was also used in this paper.

**2.3 Quantitative measures of meteorological conditions**

**2.3.1 Lower tropospheric stability**

The lower tropospheric stability (LTS) is defined as the difference in the potential temperature between 700 hPa and the surface (Slingo, 1987), and can be used to describe the thermodynamic state of the lower troposphere (Guo et al., 2016a;Guo et al., 2016b). The LTS can be used to quantitatively evaluate the vertical mixing of air pollutants in the lower troposphere:

$$LTS = \theta_{700hPa} - \theta_{surface} \tag{1}$$

A large LTS represents a high degree of stability in the lower troposphere and indicates the potential for the weak vertical mixing of air pollutants.

**2.3.2 The mean wind speed in the lower troposphere**

To quantitatively evaluate the horizontal dispersion of air pollutants, the mean wind speed (MWS) in the lower troposphere was defined as:

$$MWS = \frac{1}{h}\int_0^h \vec{V}(z)dz \tag{2}$$

where h is the height above the ground at 700 hPa and $\vec{V}(z)$ is the wind speed in the lower troposphere. This can be



simplified as follows:

$$\text{MWS} = \frac{1}{h}\sum_{i=1}^{n}\left[\overrightarrow{V_i}(z_i) + \overrightarrow{V_{i-1}}(z_{i-1})\right]\cdot 0.5\cdot\Delta z_i \qquad (3)$$

where n is the number of vertical layers from the ground to 700 hPa isobaric layer (including the 700 hPa isobaric
layer), $\overrightarrow{V_i}(z_i)$ is the wind speed in a vertical layer (when i=0 represents the wind speed on the ground and i=n
represents the wind speed at 700 hPa), and $\Delta z_i$ is the difference in height between the two adjacent vertical layers. A
large value of MWS suggests strong horizontal dispersion of air pollutants.
**3 Heavy air pollution events and weather conditions**
**3.1 Overview of the heavy air pollution events**

A total of ten heavy winter air pollution events occurred from 1 January 2006 to 31 December 2012 and from 1

January 2014 to 28 February 2017 in the urban agglomeration of Chengdu, Deyang, and Mianyang. Nine events were
accompanied by a low-pressure system at 700 hPa, and the low-pressure systems in eight events were dry and didn't
induce precipitation. This paper explores the impacts of dry low-pressure systems on the eight winter heavy air
pollution events (see **Table 1** for a summary of these eight events).

**Table 1** shows that there was low visibility during these eight heavy air pollution events in which particulate

matter is the primary pollutants. Six of the eight events were classified as persistent air pollution events, which are
harmful to the health of local residents (Chow et al., 2006;Langrish et al., 2012;Lim et al., 2012;Guo et al., 2016c), and
the longest duration was 10 days. Most of the heavy air pollution events had the characteristics of regional pollution,
with five pollution events occurring in multiple cities. Two heavy air pollution events (events 6 and 7) occurred during
the Spring Festival, with maximum daily mean $PM_{10}$ concentrations up to 403 and 562 μg m$^3$ on the Chinese New
Year Day. This suggests that the centralized letting off of fireworks during the traditional Chinese Spring Festival,
accompanied by poor conditions for the dispersion of air pollution, may lead to a sharp increase in the concentration of
particulate pollutants near ground level within a short period of time (Liao et al., 2017;Wang et al., 2007;Shi et al.,



2011;Huang et al., 2012).

**3.2 Weather systems and meteorological conditions during heavy air pollution events**

An analysis of the synoptic conditions showed that the urban agglomeration was affected by low-pressure systems
(low vortex or low trough) at 700 hPa during periods of deteriorating air quality in the eight heavy air pollution events
(**Fig. 2**). These studied areas were all located in front of low-pressure systems and were controlled by a southerly warm
air flow (**Fig. 2**). Weather systems can be characterized by their relative vorticity. A positive relative vorticity usually
corresponds to a low-pressure system, whereas a negative relative vorticity usually represents a high-pressure system.
Thus the relative vorticity at 700 hPa was analyzed during periods of both deteriorating and improving air quality
(**Table 2**).
**Table 2** shows that the relative vorticities at 700 hPa during periods of deteriorating air quality were all positive.
This indicated that the study areas were located in front of low-pressure systems at 700 hPa. As a result, a southerly
warm air flow dominated at 700 hPa and led to an increase in temperature above the atmospheric boundary layer,
which increased atmospheric stability and favored the formation of an air pollution event. During periods of improving
air quality, the relative vorticities at 700 hPa of six heavy air pollution events (except for events 6 and 7) were negative,
showing that the low-pressure systems had transited across the study areas. These areas were thus controlled by a
northerly dry, cold air flow at 700 hPa. As a consequence, the temperature above the atmospheric boundary layer
decreased and the stability of the atmosphere weakened, which favored the vertical mixing of air pollutants.
To explore the impacts of low-pressure systems on the structure of the atmospheric boundary layer, the boundary
layer height during periods of deteriorating and improving air quality were analyzed for each heavy air pollution event
(**Table 3**). In most of the heavy air pollution events, the height of the boundary layer increased after the low-pressure
system had passed across the study area. However, the increase in the height of the boundary layer was not as
significant as that seen in Eastern China (Ji et al., 2012;Quan et al., 2013;He et al., 2015;Leng et al., 2016;Qu et al.,
2017) and the boundary layer heights in air pollution events 3, 4, and 6 decreased after transit of the low-pressure





system. These results show that the effects of the transit of low-pressure systems at 700 hPa on the height of the
boundary layer were weak. It is therefore difficult to explain the variations in the concentrations of air pollutants in the
study areas by considering only the meteorological conditions within the boundary layer.
Previous studies have shown that the meteorological conditions above the boundary layer should also be
considered (Slingo, 1987;Guo et al., 2016a;Guo et al., 2016b). Therefore an index of the MWS in the lower
troposphere was proposed and this index, together with the LTS of the eight heavy air pollution events, was further
investigated (**Table 3**). The differences in the potential temperature between 700 hPa and the surface during periods of
deteriorating air quality in the eight events were all ≥18.54 K and the maximum value was 29.45 K, indicating that the
lower troposphere was very stable. The MWS was ≤4.22 m s$^{-1}$ for all eight events, with a minimum of 1.91 m s$^{-1}$.
These results show that the low-pressure systems resulted in the stagnation of air in the lower troposphere. After the
low-pressure systems had transited the study area, the lower tropospheric stability significantly decreased, with a
maximum decrease in the LTS of up to −11.23 K, and the MWS increased. This showed that the arrival of a dry, cold
air flow induced by the transit of the low-pressure system significantly weakened the stability of the lower troposphere
and increased the wind speed, improving air quality.
In events 6 and 7, however, although the study areas were still located in front of the low-pressure system and the
capacity for dispersion had not yet improved, the concentrations of particulate matter began to sharply decrease before
the transit of the low-pressure system. Both of these events occurred during the Chinese Spring Festival. After the
Chinese New Year Day, the letting off of fireworks stopped and the emission of air pollutants was significantly
reduced, resulting in a sharp decrease in the concentration of particulate matter (Wang et al., 2007;Shi et al., 2011;Liao
et al., 2017). The decrease in the magnitude of the daily mean concentration of $PM_{10}$ in event 7 was up to 350 μg m$^{-3}$.
These eight heavy air pollution events in the northwest Sichuan Basin can therefore be categorized into two types
based on their date of occurrence. The two heavy air pollution events (6 and 7) occurring during the Chinese Spring
Festival were categorized as Spring Festival excessive emission heavy air pollution events. The other six events



(events 1–5 and 8) were categorized as normal heavy air pollution events.

**4 Impacts of low-pressure systems on heavy air pollution events**

To further explore the mechanism involved in the formation of heavy air pollution events, with a particular
emphasis on the effect of low-pressure systems on air quality, a typical event was selected from the eight events
described in the preceding section. The variations in air quality and the dynamic and thermodynamic conditions in the
lower troposphere of the selected event were analyzed. Additionally, the impacts of Spring Festival excessive emission
on heavy air pollution events also been investigated.

**4.1 The influencing mechanism of low-pressure systems on heavy air pollution events**

Heavy air pollution event 8 occurred from 1 January 2017 to 6 January 2017 (Table 3) and the most polluted area
was Chengdu. The maximum daily mean concentrations of $PM_{2.5}$ and $PM_{10}$ occurred on 5 January 2017. The
maximum $PM_{10}$ daily mean concentration in Chengdu was up to 480 μg m$^{-3}$. The concentrations of particulate matter
increased sharply (**Fig. 3**) from 00:00 BST on 3 January 2017 to 00:00 BST on 5 January 2017 and the concentrations
of nitrogen dioxide and carbon monoxide also showed an increasing trend. Since 12:00 BST on 5 January 2017, the
concentrations of particulate matter decreased significantly (**Fig. 3**).
**Fig. 4** shows the weather maps at 700 hPa during event 8. **Fig. 4a** shows that there was no low-pressure system at
700 hPa over the urban agglomeration at 02:00 BST on 2 January and there was a dry, cold air flow from the northwest.
A low trough was generated at 700 hPa on the west side of the urban agglomeration at 14:00 BST on 2 January 2017.
This trough later developed and was enhanced, the urban agglomeration was still located at the front of the trough and
was controlled by a warm, moist air flow from the southwest until 02:00 BST on 5 January 2017 (**Fig. 4b** and **4c**). The
concentrations of particulate matter in the urban agglomeration increased sharply and the air quality deteriorated. The
trough developed further and a low vortex was formed, which transited across over the study area at 02:00 BST on 5
January 2017 (**Fig. 4d**). The urban agglomeration was then located behind the low vortex and was controlled by a
northerly dry, cold air flow (**Fig. 4d**) and the air pollutants were rapidly dispersed.



The west–east vertical cross-sections of the 24-hour change in temperature and wind vectors (u and w) in the most polluted area (30.75 °N) (**Fig. 5**) and the vertical profiles of temperature and horizontal wind speed (**Fig. 6**) were analyzed to investigate the effects of the low-pressure system on the dynamic and thermodynamic dispersion of air pollutants in the lower troposphere.

**Fig.4b** and **4c** shows that the urban agglomeration was located in front of the low-pressure system and was controlled by a southerly warm air flow. There was a descending motion between the top of the boundary layer and 500 hPa (**Fig. 5a** and **5b**). Under the effects of warm advection and descending motion, a warming center appeared between 800 and 650 hPa (**Fig. 5a–c**) and the maximum increase in the 24-hour temperature was up to 10 ℃ (**Fig. 6a**). Weak cooling occurred below 800 hPa, a strong temperature inversion appeared between 775 and 650 hPa (**Fig. 6a**), and the stability of the lower troposphere increased. The urban agglomeration was dominated by the low-pressure system for a long time and a long-lasting strong temperature inversion was therefore induced and maintained above the boundary layer. This was different from the temperature inversion that is often seen within the boundary layer in Eastern China (Ji et al., 2012;Li et al., 2012;Wang et al., 2014;Li and Chan, 2016;Zhang and Niu, 2016). The temperature inversion acted as a lid over the boundary layer, suppressing the dispersion of air pollutants. This lid effect restrained vertical mixing in the atmosphere and the local secondary circulation was therefore confined in the boundary layer, with its center located at about 850 hPa (**Fig. 5a–c**). The horizontal wind speed below 800 hPa was $\leq 2$ m s$^{-1}$ (**Fig. 6b**). These results indicate that vertical mixing and horizontal dispersion were weak, causing accumulation of air pollutants at the ground level. The concentrations of particulate matter then sharply increased to their peak value (**Fig. 3**), generating a heavy air pollution event.

A low vortex and trough at 700 hPa transited across the urban agglomeration and a northwestly dry, cold air flow prevailed (**Fig. 4d**). Under the influence of the cold air flow, a cooling center appeared between 800 and 650 hPa (**Fig. 5d**), whereas the air temperature increased below 800 hPa (**Fig. 5d**). As a result, the stability in the lower troposphere was weakened and the strong inversion layer gradually disappeared (**Fig. 6a**). The lid effect above the boundary layer



also disappeared, resulting in an increase in the local secondary circulation, the center of which was uplifted to 700
hPa (**Fig. 5d**). The horizontal wind speed below 800 hPa also increased (**Fig. 6b**). The air pollutants were able to
disperse over a larger space and the vertical mixing and horizontal dispersion were significantly improved. The air
quality improved and the heavy air pollution event ended.
To verify whether the mechanism involved in the formation of event 8 is used for the others heavy air pollution
events, the vertical profiles of temperature and horizontal wind speed in events 1-7 (**Fig. 7**) were explored during the
periods of both the low-pressure system controlling and transited over this urban agglomeration. Similar to the event 8,
a strong temperature inversion appeared over the study area between 800 and 650 hPa (**Fig. 7a**) when the urban
agglomeration was located in the front of low-pressure system and was controlled by a southerly warm air flow at 700
hPa. Meanwhile, the horizontal wind speed was low below 800 hPa; the wind speed at all levels below 850 hPa was ≤2
m s$^{-1}$ (**Fig. 7c**). After the low-pressure system had transited across the urban agglomeration, the strong inversion layer
above the boundary layer gradually disappeared (**Fig. 7b**), and the horizontal wind speed in the lower troposphere
increased (**Fig. 7d**). Therefore, the influencing mechanism of low-pressure system on heavy air pollution events is
common in this urban agglomeration.
**4.2 Impacts of Spring Festival excessive emission on heavy air pollution events**
**Table 1** shows that events 6 and 7 occurred during the Chinese Spring Festival when the concentration of
particulate matter increased sharply. Low concentrations of gaseous pollutants were found throughout these two events,
however, which may be related to a reduction in production or the shut-down of factories, as well as lower numbers of
vehicles during the week-long Spring Festival (Liao et al., 2017). The centralized letting off of fireworks during the
Chinese Spring Festival played an important part in the sharp increase in the concentrations of particulate matter
(Huang et al., 2012;Liao et al., 2017;Shi et al., 2011;Wang et al., 2007). We investigated the impacts of Spring Festival
excessive emission on event 6 and 7.
It's noteworthy that the emission of air pollutants increased sharply during this period of deteriorating air quality



for event 6 and 7 due to the centralized letting off of fireworks during the Chinese Spring Festival. What's more, under
the effects of low-pressure system, the strong temperature inversion appeared above the atmospheric boundary layer
(**Fig. 7a**) and the horizontal wind speed was low below 800 hPa (**Fig. 7c**). The combination of excessive emissions
with poor dispersion conditions resulted in the maximum daily concentrations of $PM_{10}$ occurring on the Chinese New
Year Day (**Table 1**). The maximum daily mean $PM_{10}$ concentration of eight heavy air pollution events occurred in
event 7 and was up to 562 μg m$^{-3}$ (**Table 1**). This shows that the excessive emissions during the short Chinese Spring
Festival were able to increase the peak concentrations of particulate matter.

Unlike in the normal heavy air pollution events, the concentrations of particulate matter began to decrease sharply

in event 6 and 7 before the low-pressure system transited over the urban agglomeration (**Fig. 8a** and **8b**), when the
strong temperature inversion was still present above the atmospheric boundary layer (**Fig. 10**) and the local secondary
circulation was still confined in the atmospheric boundary layer (**Fig. 9a** and **9b**). This indicated that these events were
strongly dependent on emissions. Thus, the centralized letting off of fireworks in the Chinese Spring Festival
combined with the impacts of low-pressure system were the main causes of these two events in this region of China.
**5 Conclusions and discussions**

We investigated the relationships between low-pressure systems and winter heavy air pollution events in the

urban agglomeration of Chengdu, Deyang, and Mianyang in the northwest Sichuan Basin and explored the influence of
dry and cold low-pressure systems on winter air quality.

A total of ten heavy winter air pollution events occurred in the urban agglomeration from 1 January 2006 to 31

December 2012 and from 1 January 2014 to 28 February 2017. The meteorological causes of eight of these air
pollution events were attributed to dry low-pressure systems (trough and low vortex) at 700 hPa. The schematic
diagram in **Fig. 11** shows that a strong temperature inversion appeared above the atmospheric boundary layer because
the urban agglomeration was located in front of low-pressure system at 700 hPa and was controlled by a warm
southerly air flow. This strong inversion layer acted as a lid over the boundary layer and suppressed the dispersion of



air pollutants, confining the local secondary circulation within the atmospheric boundary layer. The horizontal wind
speed in the lower troposphere was low. As a result, the space available for the vertical and horizontal dispersion of air
pollutants was small. The concentrations of air pollutants increased to their peak values, resulting in heavy air
pollution events.
After the low-pressure system had transited across the urban agglomeration, the strong inversion layer above the
boundary layer gradually disappeared, resulting in an increase and uplift of the secondary circulation and an increase in
the horizontal wind speed in the lower troposphere. The space available for the vertical and horizontal dispersion of air
pollutants increased and the concentrations of air pollutants decreased sharply, ending the heavy air pollution event.
The centralized letting off of fireworks during the Chinese Spring Festival was one of the main causes of the heavy air
pollution events in this region of China.
The urban agglomeration studied here, which is flanked by the eastern slopes of the Tibetan Plateau, is sensitive
to low-pressure systems moving east from the plateau (Feng et al., 2016). The complex terrain forms local secondary
circulations, which have a significant impact on air quality (Liu et al., 2009;Chen et al., 2009;Miao et al., 2015). We
found that the intensity and altitude of the local secondary circulations were markedly affected by the low-pressure
system and changes in circulation affected the local air quality. The mechanism of influence of the low-pressure
system on the local secondary circulation requires further elaboration using numerical simulation. The centralized
letting off of fireworks during the Chinese Spring Festival significantly affected the air quality (Shi et al., 2011;Huang
et al., 2012;Wang et al., 2007;Liao et al., 2017), especially during some of the heavy air pollution events in the urban
agglomeration, although the impact of fireworks on air quality was remarkably different depending on the dispersion
conditions (Li et al., 2006). Sensitivity research should therefore be carried out using models coupled with detailed
meteorological and chemical processes to quantitatively examine the impacts of the centralized emission of air
pollutants from the Chinese Spring Festival on local air quality.




## Competing interests

The authors declare that they have no conflict of interest.

## Acknowledgements

This work was supported by the National Natural Science Foundation of China (91644226, 41575138), the National Key Research Project of China-Strategy on Black Carbon Reduction and Evaluation of the Health Effects of Climate Change (2016YFA0602004), the Improvement on Competitiveness in Hiring New Faculties Fund (2013/14) of The Chinese University of Hong Kong and the Vice-Chancellor's Discretionary Fund of The Chinese University of Hong Kong (4930744). We would like to thank the following departments for the provided data, the Ministry of Environmental Protection of the People's Republic of China, the European Centre for Medium-Range Weather Forecasts, the University of Wyoming and the China Meteorological Administration.

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

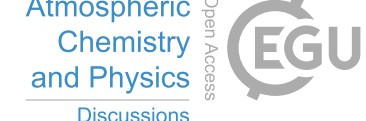

**Table 1.** Overview of the eight heavy air pollution events affected by dry low-pressure systems.

| Event | Most polluted city | Heavy air pollution event | | Most polluted day | | | End date of heavy air pollution event | | | Other cities with heavy air pollution |
| | | Start and end dates of air pollution event | $PM_{10}$ concentration range in this period ($\mu g\ m^{-3}$) | Date | $PM_{10}$ concentration ($\mu g\ m^{-3}$) | Visibility (m) | Date | $PM_{10}$ concentration ($\mu g\ m^{-3}$) | Visibility (m) | |
|---|---|---|---|---|---|---|---|---|---|---|
| 1 | Mianyang | 13–14 Jan 2006 | 284–442 | 13 Jan 2006 | 442 | 800 | 15 Jan 2006 | 166 | 12000 | Chengdu |
| 2 | Chengdu | 29 Jan 2006 | 407 | 29 Jan 2006 | 407 | <50 | 30 Jan 2006 | 190 | 11000 | None |
| 3 | Chengdu | 19–23 Dec 2006 | 348–385 | 23 Dec 2006 | 385 | 1500 | 24 Dec 2006 | 246 | 11000 | None |
| 4 | Chengdu | 21–24 Dec 2007 | 260–529 | 23 Dec 2007 | 529 | 800 | 25 Dec 2007 | 174 | 3000 | Mianyang |
| 5 | Chengdu | 18–20 Jan 2009 | 264–381 | 19 Jan 2009 | 381 | <50 | 21 Jan 2009 | 220 | 11000 | Mianyang |
| 6 | Chengdu | 3 Feb 2011 | 403 | 3 Feb 2011 | 403 | 2000 | 4 Feb 2011 | 190 | 11000 | None |
| 7 | Chengdu | 22–31 Jan 2014 | 282–562 | 31 Jan 2014 | 562 | <500 | 1 Feb 2014 | 207 | 2500 | Deyang |
| 8 | Chengdu | 1–6 Jan 2017 | 294–480 | 5 Jan 2017 | 480 | 100 | 7 Jan 2017 | 118 | 11000 | Deyang |

**Table 2.** Relative vorticity at 700 hPa during the periods of deteriorating and improving air quality in the eight heavy air pollution events.

| Event | Deteriorating air quality | | Improving air quality | |
| | Time (BST) | Relative vorticity ($1\times10^{-5}\ s^{-1}$) | Time (BST) | Relative vorticity ($1\times10^{-5}\ s^{-1}$) |
|---|---|---|---|---|
| 1 | 02:00 on 13 Jan 2006 | 2.58 | 20:00 on 13 Jan 2006 | −0.94 |
| 2 | 02:00 on 29 Jan 2006 | 4.15 | 08:00 on 30 Jan 2006 | −3.36 |
| 3 | 20:00 on 22 Dec 2006 | 4.64 | 14:00 on 23 Dec 2006 | −1.09 |
| 4 | 14:00 on 22 Dec 2007 | 0.59 | 14:00 on 23 Dec 2007 | −0.82 |
| 5 | 02:00 on 19 Jan 2009 | 1.75 | 08:00 on 19 Jan 2009 | −2.48 |
| 6 | 02:00 on 3 Feb 2011 | 2.96 | 14:00 on 3 Feb 2011 | 3.16 |
| 7 | 02:00 on 31 Jan 2014 | 9.12 | 02:00 on 1 Feb 2014 | 5.49 |
| 8 | 20:00 on 4 Jan 2017 | 6.49 | 08:00 on 5 Jan 2017 | −5.74 |





**Table 3.** Height of the atmospheric boundary layer (BLH), lower tropospheric stability (LTS), and mean wind speed (MWS) in the lower troposphere during periods of deteriorating and improving air quality in the eight heavy air pollution events.

| Event | Deteriorating air quality | | | Differences between periods of improving and deteriorating air quality | | |
|---|---|---|---|---|---|---|
| | BLH (m) | LTS (K) | MWS (m s$^{-1}$) | BLH (m) | LTS (K) | WMS (m s$^{-1}$) |
| 1 | 278.16 | 23.13 | 2.86 | 144.75 | −11.23 | 0.41 |
| 2 | 375.42 | 29.45 | 4.12 | 139.08 | −10.2 | 1.93 |
| 3 | 279.50 | 18.54 | 2.99 | −16.45 | −5.61 | 0.34 |
| 4 | 282.61 | 18.58 | 1.91 | −39.62 | −7.23 | 1.04 |
| 5 | 251.53 | 19.63 | 3.11 | 51.17 | −7.88 | 0.85 |
| 6 | 282.16 | 25.80 | 4.22 | −16.87 | 0.55 | 1.91 |
| 7 | 232.57 | 25.95 | 4.21 | 30.77 | −1.97 | −1.07 |
| 8 | 266.23 | 18.88 | 2.59 | 107.57 | −8.4 | 0.27 |

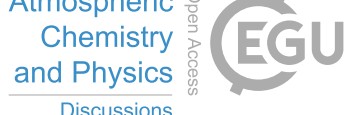


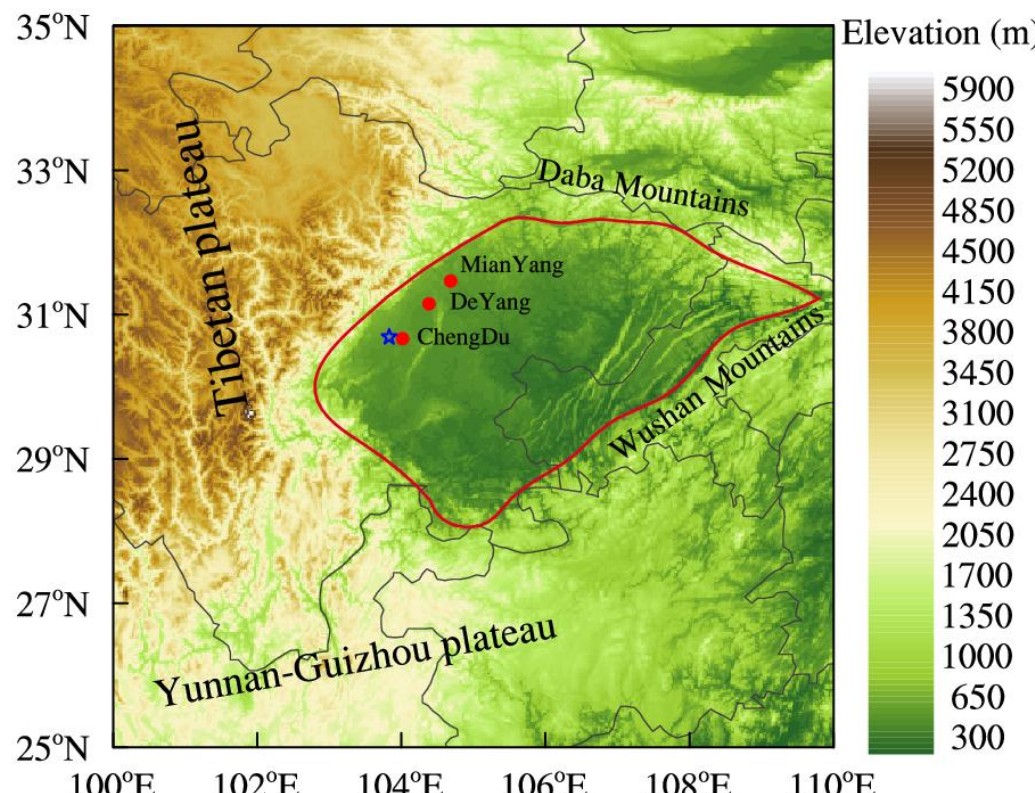


**Fig. 1** Topographic map (shading, units: m) of the Sichuan Basin (delineated in red) and surrounding areas showing the location of the cities of Chengdu, Deyang, and Mianyang (red dots). The Wenjiang station is marked with blue five-pointed stars. For interpretation of the colors, see web version of this article.





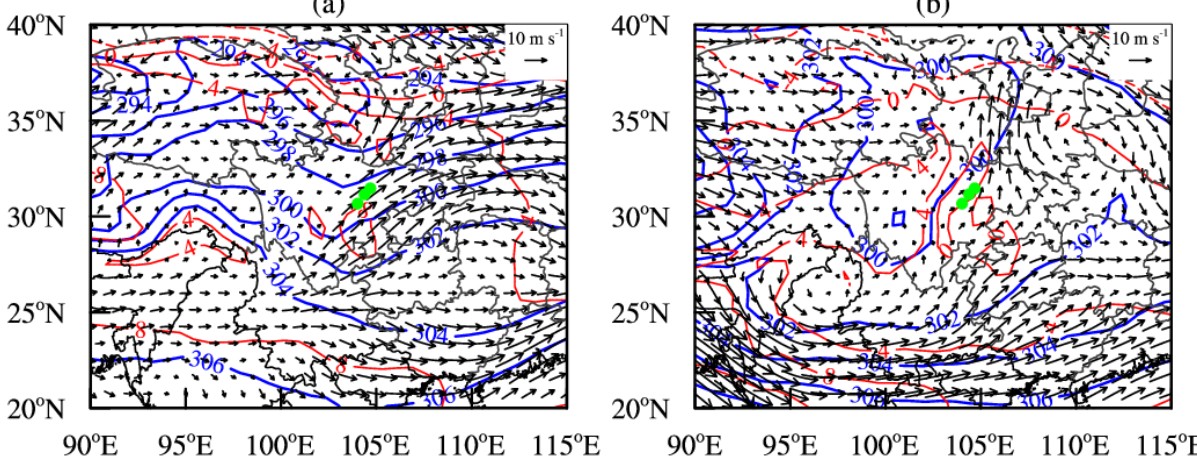


**Fig. 2** Weather maps at 700 hPa based on ERA-Interim daily data showing (a) a trough and (b) a low vortex. The blue lines are isopleths of geopotential height, the red lines are isotherms and the black arrows are wind vectors. The green dots show the location of the urban agglomeration.







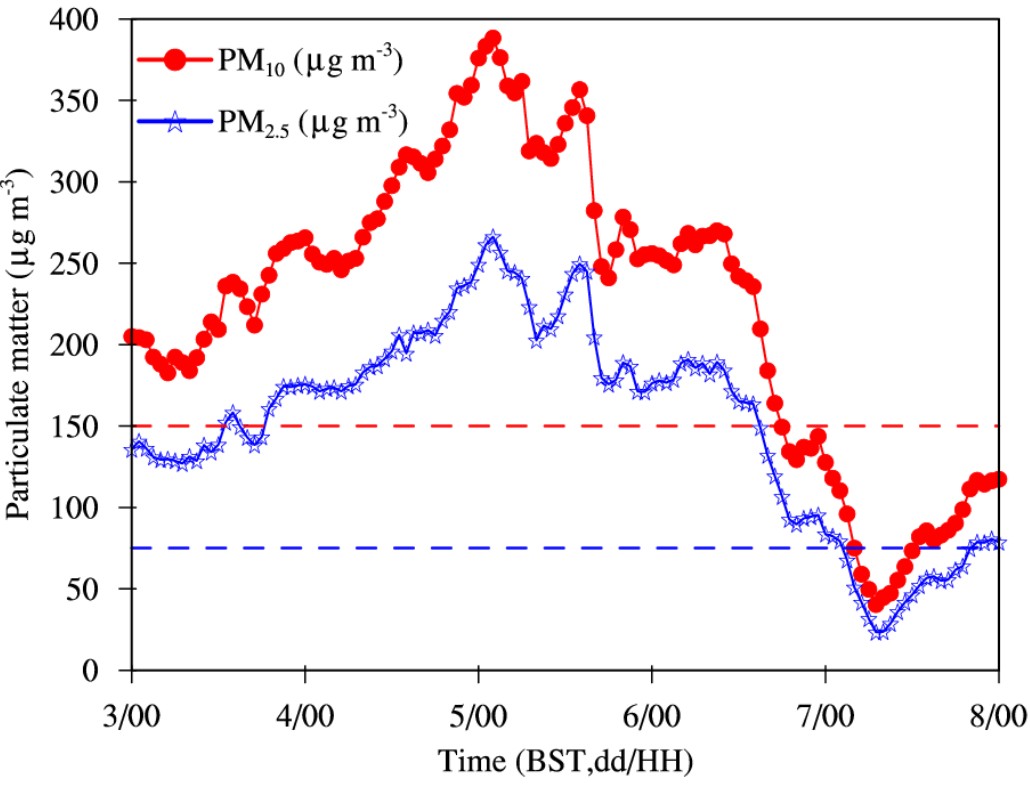


**Fig. 3** Average hourly concentrations of surface $PM_{10}$ (red solid line) and $PM_{2.5}$ (blue solid line) in the urban
agglomeration from 00:00 BST on 3 January 2017 to 00:00 BST on 8 January 2017 during event 8. The dashed red
line represents Grade II standard of $PM_{10}$ daily concentration (150 μg m$^{-3}$), the dashed blue line represents Grade II
standard of $PM_{2.5}$ daily concentration (75 μg m$^{-3}$).

562



563

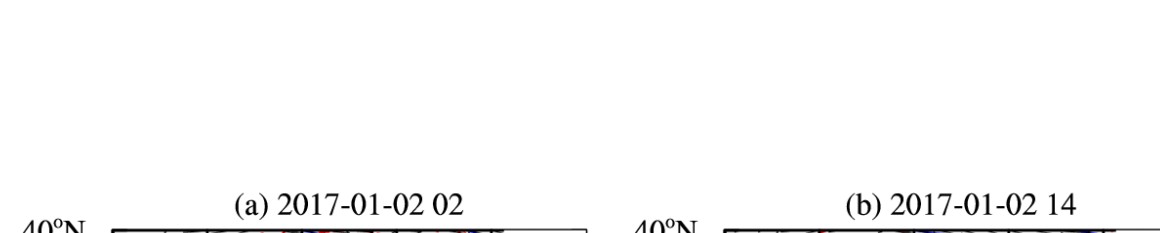

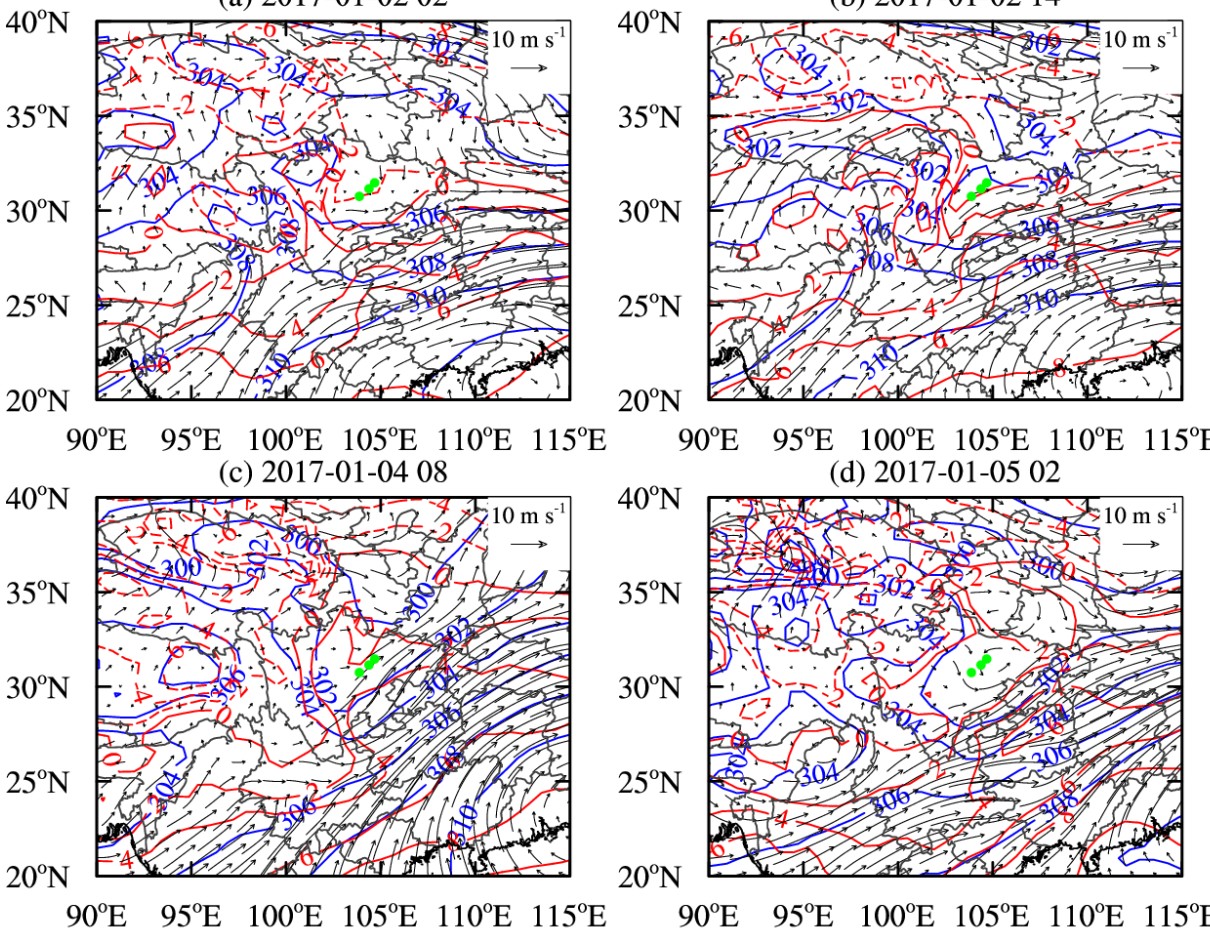

564

**Fig. 4** Weather maps at 700 hPa for event 8 at (a) 02:00 BST on 2 January 2017, (b) 14:00 BST on 2 January 2017, (c) 08:00 BST on 4 January 2017 and (d) 02:00 BST on 5 January 2017. The blue lines are isopleths of geopotential height, the red lines are isotherms and the black arrows are wind vectors. The green dots show the location of the urban agglomeration.





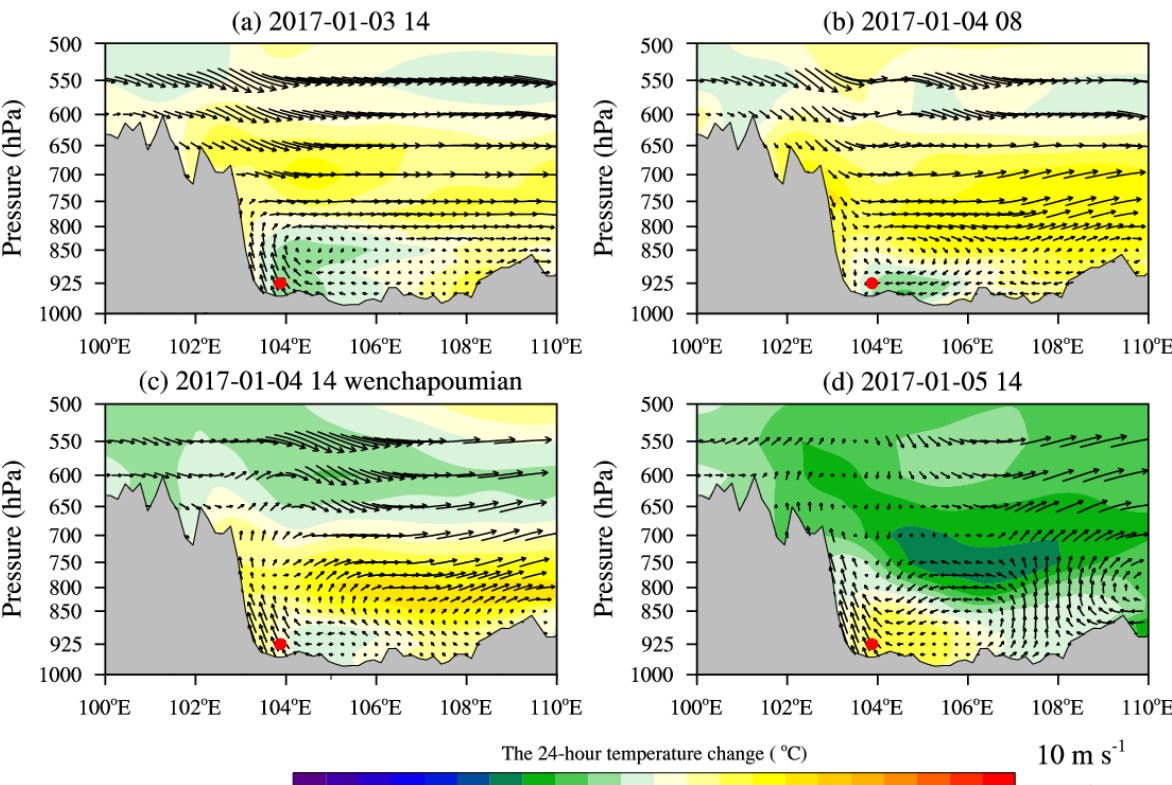


**Fig. 5** West-to-east vertical cross-sections of 24-hour temperature change (shading, units: ℃) and wind vectors
(synthesized by u and w) through the most polluted area (30.75 °N) during event 8 at (a) 14:00 BST on 3 January 2017,
(b) 08:00 BST on 4 January 2017, (c) 14:00 BST on 4 January 2017 and (d) 14:00 BST on 5 January 2017 during
event 8. Note that the vertical velocity is multiplied by 100 when plotting the wind vectors. The most polluted area is
marked by red solid dots. The gray shading represents the terrain.




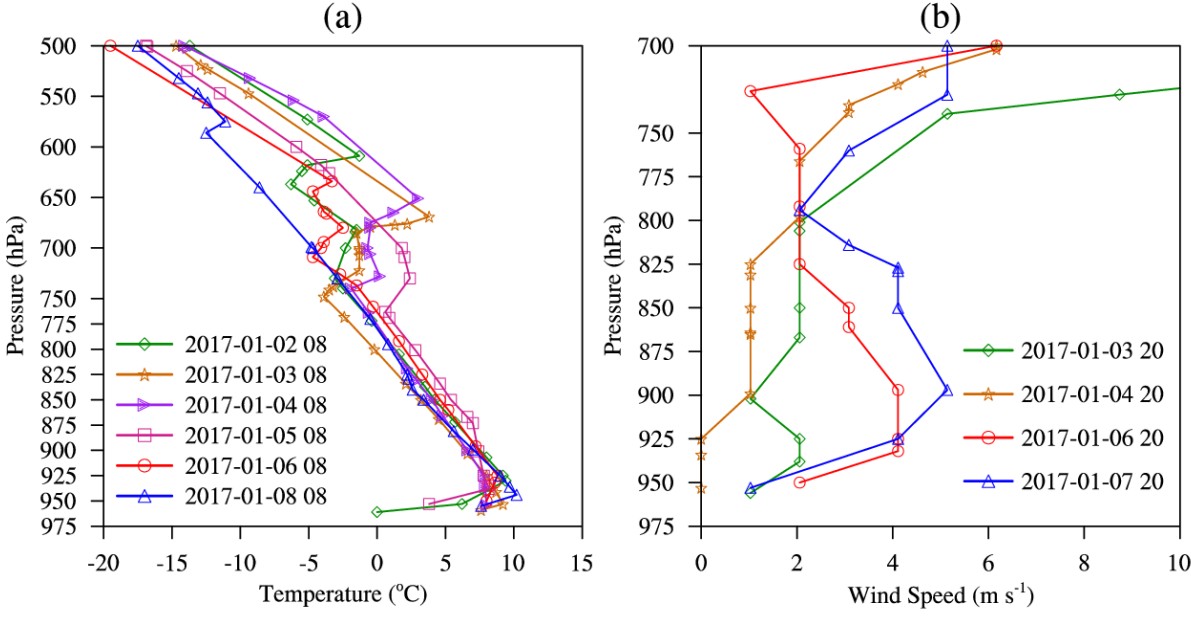


**Fig. 6** Vertical profiles of (a) temperature and (b) horizontal wind speed at Wenjiang station (30.75 °N, 103.875 °E,
see **Fig. 1**) measured by radiosonde during event 8.




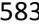





**Fig. 7** Vertical profiles of (a) temperature and (c) horizontal wind speed in the urban agglomeration during periods controlled by the low-pressure system. Vertical profiles of (b) temperature and (d) horizontal wind speed after the low-pressure system had transited across the urban agglomeration for seven heavy air pollution events (events 1–7).





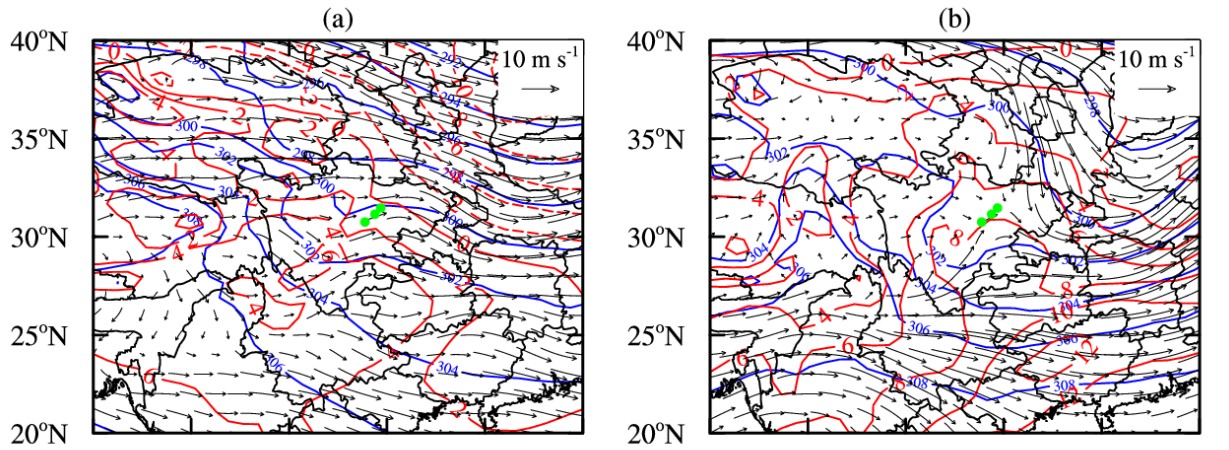


**Fig. 8** Weather maps at 700 hPa during periods of improving air quality (a) for event 6 and (b) for event 7. The blue
lines are isopleths of geopotential height, the red lines are isotherms and the black arrows are wind vectors. The green
dots show the location of the urban agglomeration.






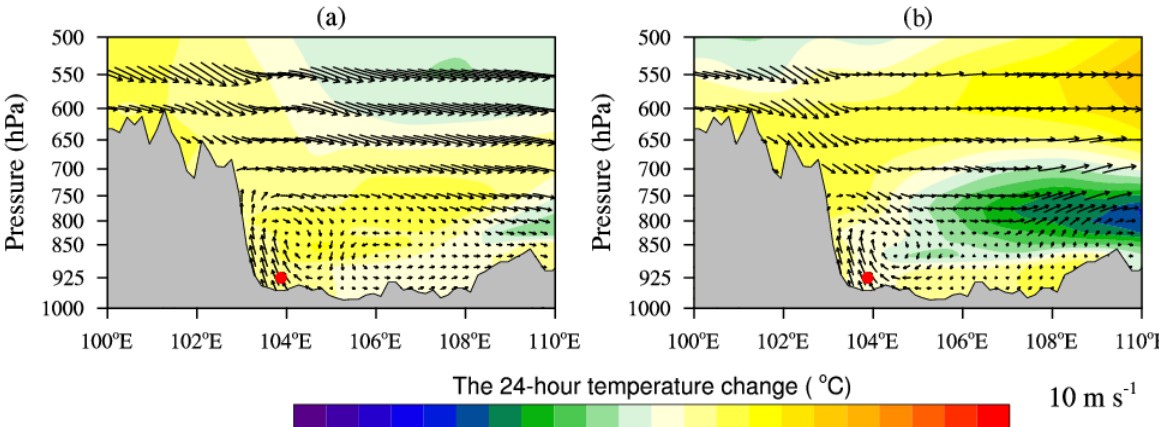


**Fig. 9** West-to-east vertical cross-sections of 24-hour temperature change (shading, units: ℃) and wind vectors (synthesized by u and w) through the most polluted area (30.75 °N) during the periods of improving air quality (a) for event 6 and (b) for event 7. Note that the vertical velocity is multiplied by 100 when plotting the wind vectors. The most polluted area is marked by red solid dots. The gray shading represents the terrain.





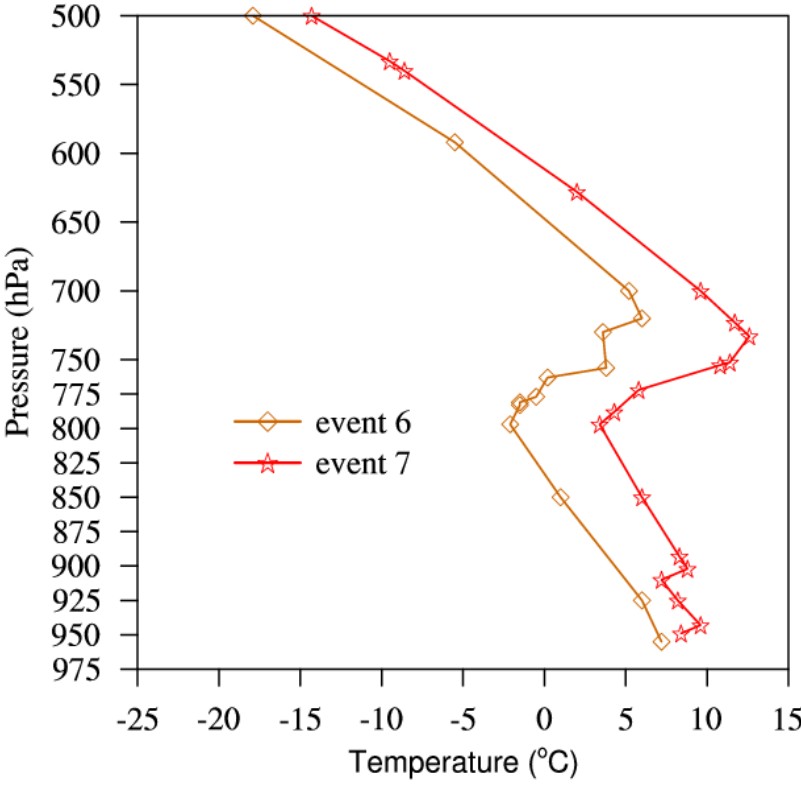


**Fig. 10** Vertical profiles of temperature at Wenjiang station (30.75 °N, 103.875 °E) measured by radiosonde during
periods of improving air quality for event 6 and 7.





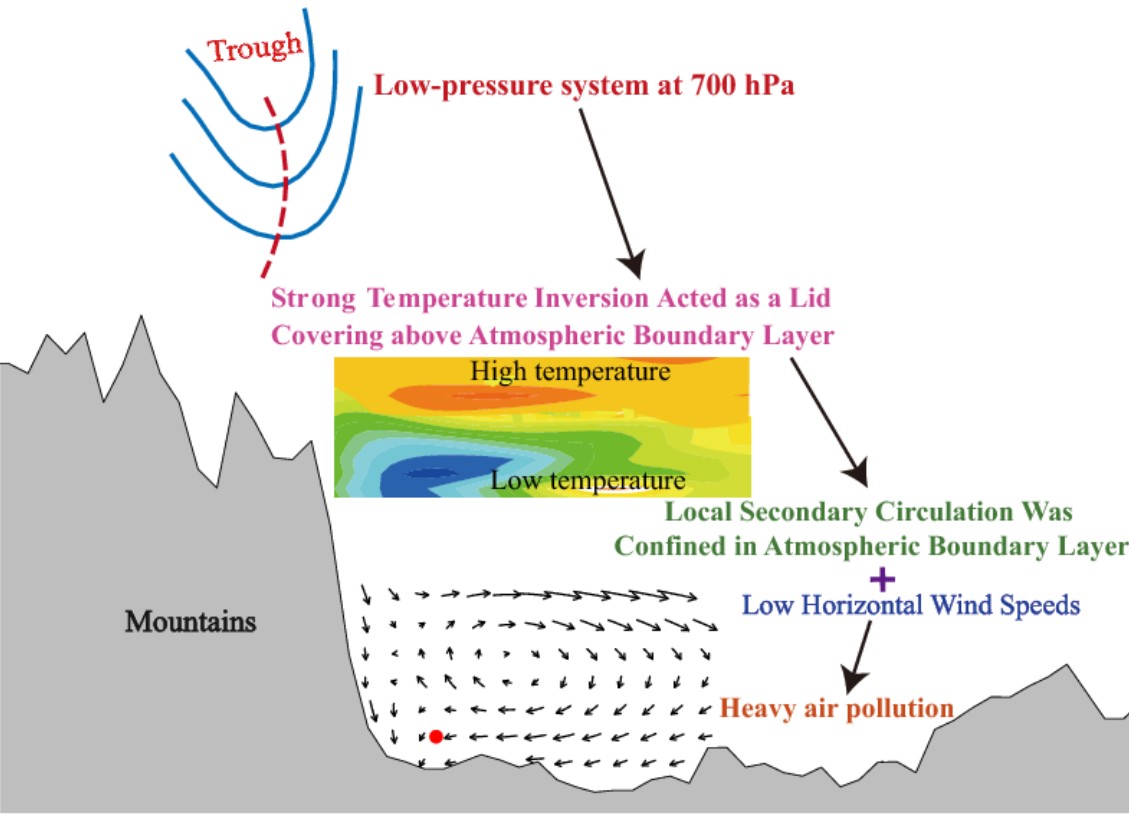

**Fig. 11** Schematic diagram of the mechanism of influence of a dry low-pressure system on winter heavy air pollution
events in the urban agglomeration.