# Peer review of "Impact of low-pressure systems on winter heavy air pollution in the northwest Sichuan Basin, China"

_Atmospheric Chemistry and Physics, 2018_

## Referee Comment (RC1) · Anonymous Referee #1 · 19 Mar 2018

This paper conducted the weather analysis of heavy PM10 pollution events in Chengdu, Deyang, and Mianyang in the northwest Sichuan Basin. Authors extracted major weather patterns, including winds, air temperature, BLH, and pressure system during the occurrence of heavy pollution in this region. The Sichuan Basin is one of several heavily contaminated regions across China and has a typical geographic terrain and persistent weather system. It is necessary to summarize the influences of such the typical terrain and weather system on air pollution prediction in the Basin. To be published in ACP, the paper needs to be improved by addressing following points.

1. From my understanding, authors used measured met data in their weather analysis.

[Figure]

They highlighted a dry low-pressure system at 700 mb, a warm southerly wind flow, and temperature inversion above the ABL as favorable weather pattern contributing to heavy pollution in their study area. A question might be raised: what is the background weather pattern in Sichuan Basin? Perhaps a better way to present their analysis is to show anomalies of these met variables from their respective long-term means during the deteriorating and improving air quality, instead of real-time measurements, such as figures 2, 4 ,5, 8, 9, etc. For example, many readers might not understand what fig. 2 is all about because we cannot figure out that wind vectors in this figure are not prevailing winds vectors and if geopotential heights represent the background GH. 2. Likewise, Table 2 presents relative vorticity at 700 hPa showing positive in deteriorating air quality but seems not telling readers how these relative vorticities were calculated. Are these departure from the mean averaged over all deteriorating and improving air quality events? Similarly, how were positive and negative BLH, LTS, and MWS in Table 3 estimated? 3. Authors constructed an index based on the results presented in Table 3 to predict the occurrence of heavy air pollution. To demonstrate the usefulness of this index, authors need to apply this index to several independent pollution events and see if the index could successfully forecast heavy pollution in the study area. 4. Discussions on Figs. 4 and 8. Discussions and interpretations of these two figures could be improved by clearly describing the lifespan of the low-pressure system and other met conditions during the pollution event. For instance, Fig. 4a shows the beginning of weather pattern causing air pollution and Fig. 4d illustrates the met conditions in the end of pollution event. As aforementioned, try to present anomalies rather than real-time data.

Other comments:

Line 123, visibility, how is visibility measured? I don't think visibility helps discussions.

Line 143, not clear wind speed on the ground. In terms of no-slip condition, wind speed at the ground surface is zero. Or the wind speed at 10 m height? How many levels from the ground surface to 700 mb? V with an upper arrow is wind vector. If Eq 3 denotes

wind speed, this upper arrow should be removed.

Line 154, any criteria being used to define a "persistent" pollution event?

Line 172, in front of low-pressure, better say east or west of the low-pressure system
Line 214, "being" should be "were"

---

## Referee Comment (RC2) · Anonymous Referee #2 · 3 Jun 2018

General comments:

Recently, air pollution issues loom large in most parts of China with the development of the economy. Sichuan Basin is one of the seriously polluted areas. This manuscript analyze the relationships between low-pressure systems and heavy air pollution events, and discuss the physical mechanisms of the heavy air pollution in winter in Sichuan basic. The ten heavy air pollution cases were used to analyse over urban agglomeration during 2006-2012 and 2014-2017 in winter, and the eight of those heavy air pollution cases were affected by a dry low-pressure system at 700hpa. When the urban agglomeration is located in front of the low-pressure system, the weather system

is controlled by the warm south wind current, and the unstable condition appears at the top of the boundary layer at the same time. The results will be helpful to improve our understanding on environment studies and fall well within the scope of ACP. The minor revisions on the present manuscript are needed before it can be published as followings:

Minor comments:

1. (P.4) Line 120-122: Why the daily average of PM10 is from the last noon to this noon during 2006-2012, but from the last midnight to this midnight during 2014-2017? Please try to describe the purpose. 2. Fig.2: What time is the result in Fig.2? 3. (P8) Line 218: from CASE 3, CASE 4, and CASE 5, the results that is the effect of the low pressure system at 700hpa causing the value of Boundary Layer height fall. Please describe the reasonableness. We know, the inversion disappears at the higher level, the wind speed increases in the lower layer, the turbulent motion enhancement, and the boundary layer height increases in the boundary layer when the low-pressure system at 700hPa passed. 4. Table 3, Please add instructions on how to calculate the boundary layer height. The values in the table 3 are average results, rightïïj§ 5. CASE 6, the whole pollution process lasts a day, but the relative vorticity of air quality is 02: 00 on February 3, but the air quality improvement is 14: 00 on February 3 in Table2. Please confirm the reasonableness of the boundary layer height. 6. CASE 6 and 7, the low-pressure system at 700hPa throughout all the pollution process, the value of pollutant concentration was decreased quickly, why? due to fireworks only? are other processes affecting pollution ? 7. Fig.6, some discussions about the evolution of the PBL height may be also good for a more complete picture. 8. CASE 6 and CASE 7, the stronger wind shear at 850hPa means the stronger dynamic turbulence (Fig. 9). How about the characteristics of the wind profile in the boundary layer (refer to Table 3) ? 9. Please unify the format of the references, such as uppercase and lowercase.

---

## Author Comment (AC1) · 5 Jul 2018

We would like to thank the referee for his/her valuable comments and suggestions which help us improve the quality of the manuscript. All the comments and concerns raised by the referee have been answered carefully point-by point as below and the corresponding parts in the manuscript have been improved.

The original comments are copied here in black color.

Author's responses are in blue color.

All changes to the manuscript have been highlighted with red color in the submitted

revised manuscript.

This paper conducted the weather analysis of heavy PM10 pollution events in Chengdu, Deyang, and Mianyang in the northwest Sichuan Basin. Authors extracted major weather patterns, including winds, air temperature, BLH, and pressure system during the occurrence of heavy pollution in this region. The Sichuan Basin is one of several heavily contaminated regions across China and has a typical geographic terrain and persistent weather system. It is necessary to summarize the influences of such the typical terrain and weather system on air pollution prediction in the Basin. To be published in ACP, the paper needs to be improved by addressing following points.

Response: Thank you very much for your positive comments and nice summary.

General comments

1. From my understanding, authors used measured met data in their weather analysis. They highlighted a dry low-pressure system at 700 mb, a warm southerly wind flow, and temperature inversion above the ABL as favorable weather pattern contributing to heavy pollution in their study area. A question might be rasied: what is the background weather pattern in Sichuan Basin? Perhaps a better way to present their analysis is to show anomalies of these met variables from their respective long-term means during the deteriorating and improving air quality, instead of real-time measurements, such as figures 2, 4, 5, 8, 9, etc. For example, many readers might not understand what fig.2 is all about because we cannot figure out that wind vectors in this figure are not prevailing winds vectors and if geopotential heights represent the background GH.

Response: Thank you very much for your constructive comments. In order to present our analysis in a better way, the anomalies of geopotential heights and wind vectors at 700 hPa, the anomalies of west-to-east vertical cross-section of 24-hour temperature change and wind vectors (synthesized by u and w), and the anomalies of temperature vertical profiles were analyzed in the revised manuscript.

To explore the differences between these low-pressure systems and the background of winter atmospheric circulation over there, the anomalies of wind vectors and geopotential heights at 700 hPa were calculated (Fig. S1). The calculation method is as follows: the averaged wind vectors and geopotential heights at 700 hPa during periods of deteriorating air quality in the above eight events subtracted from their winter mean values from 1 January 2006 to 31 December 2012 and from 1 January 2014 to 28 February 2017. As illustrated in Fig. S1, the anomalies of geopotential heights were negative in the northwest of the urban agglomeration during periods of deteriorating air quality in these heavy air pollution events. As a result, this urban agglomeration was located in front of an anomalous cyclone and was controlled by a strong southerly anomaly wind (Fig. S1).

Additionally, the anomalies of west-to-east vertical cross-section of 24-hour temperature change and wind vectors (synthesized by u and w) (Fig. S2), and the anomalies of temperature vertical profiles (Fig. S3) were also analyzed to further investigate the influencing mechanism of low-pressure system on heavy air pollution events. Fig. S2 shows that anomalous warming appeared above the atmospheric boundary layer, while anomalous cooling was observed within the boundary layer when the urban agglomeration was located in front of low-pressure system and was controlled by a southerly warm air flow at 700 hPa. This vertical structure of the anomalies of 24-hour temperature change led to an increase in the stability of the lower troposphere. As illustrated in Fig. S3, the positive anomalies of temperature between 1500 m and 3000 m above the ground level increased significantly with height. The maximum value of positive anomalies appeared at about 3000 m and was up to 9 °C. These features revealed that a strong temperature inversion existed above the boundary layer and suppressed the vertical exchange of atmosphere. As a result, the anomalous secondary circulation was also confined in the boundary layer, with its center located at about 925 hPa (Fig. S2). These results of anomalies analysis were consistent with the above analysis for real-time data, and further proved that the influencing mechanism of low-pressure system on heavy air pollution events is credible.

2. Likewise, Table 2 presents relative vorticity at 700 hPa showing positive in deteriorating air quality but seems not telling readers how these relative vorticities were calculated. Are these departure from the mean averaged over all deteriorating and improving air quality events? Similarly, how were positive and negative BLH, LTS, and MWS in Table 3 estimated?

Response: Thank you very much for your valuable comments. The values of relative vorticity at 700 hPa in Table 2 and the values of BLH, LTS, and MWS in Table 3 were not departure from the mean averaged over all deteriorating and improving air quality events. They all were estimated in each of the eight heavy air pollution events. It is considered as a better way to characterize the transits of low-pressure systems for each heavy air pollution event and to estimate the impacts of low-pressure systems on the dispersion capacity of air pollutants in the lower troposphere.

According to your comments, the detailed descriptions of the captions for these met variables in Table 2 and Table 3 have been revised as follows:

Table 2. Relative vorticity at 700 hPa during the periods of deteriorating and improving air quality in each of the eight heavy air pollution events.

Table 3. Height of the atmospheric boundary layer (BLH), lower tropospheric stability (LTS), and mean wind speed (MWS) in the lower troposphere during periods of deteriorating air quality in each of the eight heavy air pollution events, and the differences of them between periods of improving and deteriorating air quality in each event.

3. Authors constructed an index based on the results presented in Table 3 to predict the occurrence of heavy air pollution. To demonstrate the usefulness of this index, authors need to apply this index to several independent pollution events and see if the index could successfully forecast heavy pollution in the study area.

Response: Thank you very much for your valuable comments.

First, the index of mean wind speed (MWS) in the lower troposphere was constructed

based on the concept of ventilation coefficient, which has been widely used to measure the capability of air pollutants' dispersion in the eastern plains of China (Deng et al., 2014; Lu et al., 2012; Tang et al., 2015). Thus, the MWS has a certain physical meaning and rationality. The construction basis and specific method of MWS have been added in the revised manuscript.

Sichuan Basin belongs to a low wind speed zone in China due to its deep mountain-basin topography, and the wind speed in the mixing layer is often low and with small change magnitudes (Chen and Xie, 2012; Huang et al., 2017; Wang et al., 2018). For analyzing air quality in Sichuan Basin, the meteorological conditions in the lower troposphere that can reflect ventilation should be considered. To quantitatively evaluate the horizontal dispersion of air pollutants in Sichuan Basin, the mean wind speed (MWS) in the lower troposphere was constructed based on the concept of ventilation coefficient (VC is a product of mixing layer height multiplied by average wind speed through the mixing height). In the eastern plains of China, the ventilation coefficient has been widely used to measure the capability of air pollutants' dispersion (Deng et al., 2014; Lu et al., 2012; Tang et al., 2015).

Second, the usefulness of this new index (WMS) have been demonstrated to be good by several independent pollution events. As shown in Table 3, the values of this new index (MWS) of these six events (1–5 and 8) increased significantly after the low-pressure systems had transited the urban agglomeration, and the air quality of these six events improved significantly. For the events 6 and 7 which occurred during the Spring Festival, the improvement of their air quality was mainly attributable to the stop of the letting-off of fireworks. These results revealed that the new index could successfully forecast heavy pollution in the study area.

4. Discussions on Figs. 4 and 8. Discussions and interpretations of these two figures could be improved by clearly describing the lifespan of the low-pressure system and other met conditions during the pollution event. For instance, Fig. 4a shows the beginning of weather pattern causing air pollution and Fig. 4d illustrates the met conditions

in the end of pollution event.

Response: Thank you very much for your valuable comments. This manuscript have been revised according your comments.

Other comments

1. Line 123, visibility, how is visibility measured? I don't think visibility helps discussions.

Response: We agree with your comments, and the visibility has been removed in our revised manuscript.

2. Line 143, not clear wind speed on the ground. In terms of no-slip condition, wind speed at the ground surface is zero. Or the wind speed at 10 m height? How many levels from the ground surface to 700 mb? V with an upper arrow is wind vector. If Eq 3 denotes wind speed, this upper arrow should be removed.

Response: Thank you very much for your valuable comments. In line 143, the new index (MWS) was calculated based on sounding data which were measured at Wenjiang station (see Fig. 1) in Chengdu. Thus, "Wind speed on the ground" has been revised to "Wind speed at the ground surface". The vertical levels from the ground surface to 700 mb were not fixed. In general, the number of vertical levels was more than six. The V in Eq 3 denotes wind speed, and thus the upper arrow of V has been removed in the revised manuscript.

3. Line 154, any criteria being used to define a "persistent" pollution event?

Response: A "persistent" pollution event was defined by two or more consecutive days with daily PM10 mean concentration $\geq$ 250 $\mu$g m$-3$. Moreover, this criteria being used to define a "persistent" pollution event has been added in the revised manuscript.

A "persistent" pollution event was defined by two or more consecutive days with daily PM10 mean concentration $\geq$ 250 $\mu$g m$-3$, which is reported to be harmful to the health

of local residents (Chow et al., 2006; Guo et al., 2016c; Langrish et al., 2012; Lim et al., 2012).

4. Line 172, in front of low-pressure, better say east or west of the low-pressure system.

Response: Agreed and corrected in the revised manuscript.

5. Line 214, "being" should be "were"

Response: Agreed and corrected in the revised manuscript.

[Figure]

**Fig. 1.** Fig. S1 The anomalies of geopotential heights (shading, units: dagpm) and wind vectors (black arrows) at 700 hPa (the averaged wind vectors and geopotential heights at 700 hPa during periods of deteri

[Figure]

The anomalies of 24-hour temperature change ( °C)        5 m s⁻¹

-6  -5  -4  -3  -2  -1  0  1  2  3  4  5  6

**Fig. 2.** Fig. S2 West-to-east vertical cross-section of the anomalies of 24-hour temperature change and wind vectors (synthesized by u and w) through the most polluted area (30.75° N) (the averaged 24-hour tem

[Figure]

**Fig. 3.** Fig. S3 Vertical profiles of temperature anomalies at Wenjiang station (30.75° N, 103.875° E) measured by radiosonde (the averaged temperature during periods of deterio-rating air quality in the eight

---

## Author Comment (AC2) · 5 Jul 2018

Thank you very much for your constructive comments which help us improve the quality of the manuscript. We have carefully modified the manuscript according to your comments. We hope you will be satisfied with our revisions.

The original comments are copied here in black color.

Author's responses are in blue color.

All changes to the manuscript have been highlighted with red color in the submitted revised manuscript.

[Figure]

General comments

Recently, air pollution issues loom large in most parts of China with the development of the economy. Sichuan Basin is one of the seriously polluted areas. This manuscript analyze the relationships between low-pressure systems and heavy air pollution events, and discuss the physical mechanisms of the heavy air pollution in winter in Sichuan basic. The ten heavy air pollution cases were used to analyse over urban agglomeration during 2006-2012 and 2014-2017 in winter, and the eight of those heavy air pollution cases were affected by a dry low-pressure system at 700hPa. When the urban agglomeration is located in front of the low-pressure system, the weather system is controlled by the warm south wind current, and the unstable condition appears at the top of the boundary layer at the same time. The results will be helpful to improve our understanding on environment studies and fall well within the scope of ACP. The minor revisions on the present manuscript are needed before it can be published as followings.

Response: Thank you very much for your positive comments and nice summary.

Minor comments

1. (P.4) Line 120-122: Why the daily average of PM10 is from the last noon to this noon during 2006-2012, but from the last midnight to this midnight during 2014-2017? Please try to describe the purpose.

Response: Thank you very much for your valuable comments.

The third revision of the "Ambient Air Quality Standard" (AAQS) (GB3095-2012) in China was released on February 29th, 2012, replacing the old "Ambient Air Quality Standard" (AAQS) (GB3095-1996). This new standard (GB3095-2012) began to be carried out gradually since 2013. Thus, the daily average of PM10 was from the last noon to this noon during 2006-2012 based on the new "Ambient Air Quality Standard" (AAQS) (GB3095-2012). However, based on the old "Ambient Air Quality Standard"

(AAQS) (GB3095-1996), the daily average of PM10 was from the last midnight to this midnight during 2014-2017. These detailed descriptions have been added in the revised manuscript.

The third revision of the "Ambient Air Quality Standard" (AAQS) (GB3095-2012) was released on February 29th, 2012, replacing the old "Ambient Air Quality Standard" (AAQS) (GB3095-1996) and PM2.5 was adopted into the AAQS in China since 2013. The air quality monitoring stations needed to be updated and the data of air pollutants monitored in the three cities existed missing measurement during 2013. Thus, the winter heavy pollution events during 2013 had not been analyzed in this paper. Moreover, the PM10 daily mean concentration from 1 January 2014 to 28 February 2017 refers to the 24-hour average concentration of PM10 from 00:00 BST (Beijing Standard Time, i.e., Coordinate Universal Time (UTC) +8 h) to 24:00 BST on the current day based on the new "Ambient Air Quality Standard" (AAQS) (GB3095-2012). However, based on the old "Ambient Air Quality Standard" (AAQS) (GB3095-1996), the PM10 daily mean concentration from 1 January 2006 to 31 December 2012 refers to the 24-hour average concentration of PM10 from 12:00 BST on the previous day to 12:00 BST on the current day.

2. Fig.2: What time is the result in Fig.2?

Response: Thank you very much for this question. The weather maps at 700 hPa based on ERA-Interim daily data show Fig. 2(a) a trough from event 2 at 20:00 BST on 28 January, 2006 and Fig. 2(b) a low vortex from event 4 at 14:00 BST on 22 December, 2007. The information is added in the figure caption.

Fig. 2 Weather maps at 700 hPa based on ERA-Interim daily data showing (a) a trough from event 2 at 20:00 BST on 28 January, 2006 and (b) a low vortex from event 4 at 14:00 BST on 22 December, 2007. The blue lines are isopleths of geopotential height, the red lines are isotherms and the black arrows are wind vectors. The green dots show the location of the urban agglomeration.

3. (P8) Line 218: from CASE 3, CASE 4, and CASE 5, the results that is the effect of the low pressure system at 700 hpa causing the value of Boundary Layer height fall. Please describe the reasonableness. We know, the inversion disappears at the higher level, the wind speed increases in the lower layer, the turbulent motion enhancement, and the boundary layer height increases in the boundary layer when the low-pressure system at 700 hPa passed.

Response: Thank you very much for your valuable comments.

First, Sichuan Basin belongs to a low wind speed zone in China due to its deep mountain-basin topography (Fig. 1). The wind speed in the boundary layer is often low and with small change magnitudes (Chen and Xie, 2012; Huang et al., 2017; Wang et al., 2018), and the cold air induced by the transit of low-pressure systems usually can't reach in the ground layer (Fig. 5). As a result, the increased magnitudes of wind speed (Fig. 6b, Fig. 7 c and 7d) and the change magnitudes of temperature (Fig. 6a, Fig. 7a and 7b) were very small in the boundary layer after the low-pressure system at 700 hPa passed. Especially for events 3 and 4, the wind speed decreased and a temperature inversion formed in the boundary layer. Thus, the boundary layer heights in air pollution events 3 and 4 decreased after transit of the low-pressure system.

Second, there was a typo in the sentence of "the boundary layer heights in air pollution events 2, 4, and 6 decreased after transit of the low-pressure system". For event 6 which occurred during the Spring Festival of China, the improvement of its air quality was mainly attributable to the stop of the letting-off of fireworks. As shown in Table 2 and Table 3, the study areas were still located in the front of the low-pressure system and the capacity for dispersion had not yet improved (including the boundary layer height decreased) when the air quality started to improve. Event 6 should be therefore removed in this sentence.

Third, the detailed descriptions about the reasonableness have been added in the revised manuscript according to your comments.
From Fig.6 and Fig.7, we also found some interesting features that the effects of the transit of low-pressure systems at 700 hPa on the meteorological factors within the boundary layer were weak. These features may be related to its deep mountain-basin topography (Fig. 1). Under the effects of the deep mountain-basin topography, wind speed in the boundary layer is often low and with small change magnitudes (Chen and Xie, 2012; Huang et al., 2017; Wang et al., 2018), and cold air induced by the transit of low-pressure systems usually can't reach to the ground layer (Fig. 5). As a result, the increased magnitudes of wind speed (Fig. 6b, Fig. 7 c and 7d) and the change magnitudes of temperature (Fig. 6a, Fig. 7a and 7b) were very small in the boundary layer after the low-pressure system at 700 hPa passed. Especially for events 3 and 4, the wind speed decreased and a temperature inversion formed in the boundary layer. These characteristics of the wind and temperature profiles in the boundary layer were the key factors leading to the evolution of boundary layer height as shown in Table 3.

4. Table 3, Please add instructions on how to calculate the boundary layer height. The values in the table 3 are average results, right?

Response: Yes, the values in the Table 3 are average results. The height of atmospheric boundary layer was obtained from the ERA-Interim daily dataset at the surface with 3 h temporal resolution (00:00, 03:00, 06:00, 09:00, 12:00, 15:00, 18:00, and 21:00 UTC)(http://apps.ecmwf.int/datasets/data/interim-full-daily/levtype=sfc/). This boundary layer height was defined as the level where the bulk Richardson number, based on the difference between quantities at that level and the lowest model level, reaches the critical value $Ri_{cr} = 0.25$ (Beljaars, 2006). The instructions on how to calculate the boundary layer height have been added in the revised manuscript.

5. CASE 6, the whole pollution process lasts a day, but the relative vorticity of air quality is 02:00 on February 3, but the air quality improvement is 14: 00 on February 3 in Table 2. Please confirm the reasonableness of the boundary layer height.

Response: Thank you very much for your valuable comments. The boundary layer

height in event 6 has been confirmed to be correct according to your comments. As shown in the response to the third minor comment, in event 6, which occurred during the Spring Festival of China, the improvement of its air quality was mainly attributable to the stop of the letting-off of fireworks. As shown in Table 2 and Table 3, the study areas were still located in the front of the low-pressure system, and the capacity for dispersion has not yet improved (including the decrease in boundary layer height) when the air quality started to improve. The boundary layer height has not increased during the periods of improving air quality in event 6 because the low-pressure system has not yet passed.

6. CASE 6 and 7, the low-pressure system at 700 hPa throughout all the pollution process, the value of pollutant concentration was decreased quickly, why? due to fireworks only? are other processes affecting pollution ?

Response: Thank you very much for this constructive comment.

First, the effects of fireworks on air quality in Chengdu during Chinese New Year (CNY) from 2013 to 2017 have been investigated. The results showed that time-variations in PM10 concentration during CNY were similar in these five years, even though their meteorological conditions were different. As illustrated in Fig. S4, PM10 concentration increased sharply during the periods of the letting-off of fireworks in CNY, and began to decrease significantly after the letting-off of fireworks stopped. These results were consistent with the changes in particulate pollutant concentrations during CNY in other cities of China (http://www.zhb.gov.cn/gkml/hbb/qt/201702/t20170201_395336.htm). It is a common phenomenon that PM10 concentrations decreased sharply after the letting-off of fireworks stopped during CNY. Additionally, to evaluate the effects of excessive emission about fireworks on air quality in a better way, we analyzed the diurnal variations of the differences of averaged PM10 concentration in Chengdu between during in the periods of the letting-off of fireworks in CNY (defined as the period from 12:00 BST on the Eve of CNY to 12:00 BST on 1 Lunar January) and 5 days before the letting-off fireworks, and between during 5 days after the letting-off of fireworks in CNY and in

the periods of the letting-off of fireworks from 2013 to 2017, see Fig. S5. The letting-off of fireworks during CNY was observed to have a significant effect on the air quality in Chengdu. Especially during 5 days after the letting-off of fireworks stopped, production was reduced, factories were shut-down and the numbers of vehicles were lower due to the week-long holiday of CNY (Liao et al., 2017). As a result, the maximum decrease in the magnitude of PM10 concentration was more than 220 $\mu$g m-3 and occurred at night from 00:00 BST to 06:00 BST (Fig. S5) which corresponded to the period of the centralized letting-off of fireworks.

Second, unlike in the normal heavy air pollution events, the concentrations of particulate matter began to decrease sharply in event 6 and 7 before the low-pressure system transited over the urban agglomeration (Fig. 8), when the strong temperature inversion still existed above the boundary layer (Fig. 10), the local secondary circulation was still confined in the boundary layer (Fig.9) and the capacity for dispersion has not yet improved significantly (Table 3).

Based on the above analysis results, we conclude that the sharp decreases in PM10 concentration for event 6 and 7 were mainly attributable to the significant reduction in emissions induced by the letting-off of fireworks stopped and the week-long holiday of CNY. The detailed discussions had been added in the revised manuscript.

7. Fig.6, some discussions about the evolution of the PBL height may be also good for a more complete picture.

Response: Thank you very much for this valuable comment. According to your comments, in-depth discussions of Fig. 6 and Fig. 7 were added to explain the evolution of PBL height. The detailed discussions had been made in the response to the third minor comment.

8. CASE 6 and CASE 7, the stronger wind shear at 850 hPa means the stronger dynamic turbulence (Fig. 9). How about the characteristics of the wind profile in the boundary layer (refer to Table 3) ?

Response: As shown in the response to the third minor comment, wind speeds in the boundary layer is often low and with small change magnitudes (Fig. 6b, Fig. 7 c and 7d). In order to explain the evolution of PBL height in Table 3, the characteristics of the wind profile in the boundary layer have been analyzed and added in the revised manuscript.

9. Please unify the format of the references, such as uppercase and lowercase.

Response: the format of the references have been unified according to your comments.

References

Beljaars, A.: Chapter 3: Turbulent transport and interactions with the surface. Part IV: Physical Processes, IFS Documentation, Operational implementation 12 September 2006 Cy31r1 31, ECMWF, Shinfield Park[J], Reading, RG2 9AX, England, 2006.

Chen, Y., and Xie, S.: Temporal and spatial visibility trends in the Sichuan Basin, China, 1973 to 2010[J], Atmos. Res., 112, 25-34, https://doi.org/10.1016/j.atmosres.2012.04.009, 2012.

Huang, Q., Cai, X., Song, Y., and Zhu, T.: Air stagnation in China (1985–2014): climatological mean features and trends[J], Atmos. Chem. Phys., 17, 7793-7805, https://doi.org/10.5194/acp-17-7793-2017, 2017.

Liao, T., Wang, S., Ai, J., Gui, K., Duan, B., Zhao, Q., Zhang, X., Jiang, W., and Sun, Y.: Heavy pollution episodes, transport pathways and potential sources of PM2.5 during the winter of 2013 in Chengdu (China)[J], Sci. Total Environ., 584-585, 1056-1065, https://doi.org/10.1016/j.scitotenv.2017.01.160, 2017.

Wang, X., Dickinson, R. E., Su, L., Zhou, C., and Wang, K.: PM2.5 pollution in China and how it has been exacerbated by terrain and meteorological conditions[J], Bull. Am. Meteorol. Soc., 99, 105-119, http://dx.doi.org/10.1175/BAMS-D-16-0301.1, 2018.

Please also note the supplement to this comment:

[Figure]

https://www.atmos-chem-phys-discuss.net/acp-2018-61/acp-2018-61-AC2-supplement.pdf
* * *
[Figure]

[Figure]

**Fig. 1.** Fig. S4 The hourly concentrations of PM10 during Chinese New Year (CNY) in 2014 for event 7 (red solid line) and the averaged PM10 concentrations during CNY in five years from 2013 to 2017 (blue solid

[Figure]

**Fig. 2.** Fig. S5 The diurnal variations of the differences of averaged PM10 concentration in Chengdu between during in the periods of the letting-off of fireworks in CNY and 5 days before the letting-off firew

---

## Author Response (AR2)

Response to Anonymous Referee #1

Thanks the reviewer for further reviewing our manuscript. All the comments raised by the reviewer have been answered carefully point-by point as below and the corresponding parts in the manuscript have been improved.

The original comments are copied here in black color.

Author's responses are in blue color.

All changes to the manuscript have been highlighted with red color in the submitted revised manuscript.

The revised paper addressed the major points in my previous comments. I recommend accept for the publication in ACP after further minor revisions as outlined below.

1. Line 145, "with small change magnitude" rewritten as "less changed"

   Response: Corrected (Line 141).

2. Line 146-147, "the meteorological conditions … considered" changed to "the meteorological conditions contributing to the atmospheric ventilation should be taken into consideration"

   Response: Above mentioned suggestion has been corrected in manuscript at line 143-144

3. Line 151, delete "However"

   Response: Corrected (Line 148)

4. Line 152-154, "To quantitatively … coefficient" repeat above sentence, please delete it.

   Response: Thank you very much for your valuable comments. This sentence has been deleted (Line 148-151).

5. Line 188, "the anomalies…", better written as "composite anomalies…". To my understanding, composite anomalies were the departure of the mean wind vectors and GH at 700 hPa averaged over the eight deteriorating air quality events from their respective means from Jan 1 2006 to Dec 31 2012 and from Jan 1 2014 to Feb 28 2017

   Response: Agreed and corrected in manuscript at line 185-186.

6. Line 188, delete "over there"

Response: Corrected (Line 185)

7. Line 253, delete "Soon", rewrite sentence as "As shown in Fig. 4b, a low trough was subsequently generated…"

Response: Corrected (Line 251)

8. Line 300, "composite anomalies"

Response: Agreed and corrected in manuscript at line 298 and all similar expressions in the supplement have also been revised.

9. Line 305, temperature change or temperature? Below you said "temperature"

Response: Represents "temperature change". In order to avoid the ambiguity, the related instruction has been added in manuscript at line 304.

10. Line 313-314, "that the" changed to "characterized by the weak effects…", delete "were weak"

Response: Agreed and corrected in manuscript at line 312-313.

11. Line 316, "with small change magnitude" change to "less altered"

Response: Agreed and corrected in manuscript at line 315-316.

12. Line 318, delete "magnitudes of"

Response: Corrected (Line 318).

13. Line 318-319, "the change of magnitudes of temperature" changed to "the degree of change in temperature"

Response: Corrected (Line 318-319).

14. Line 341, delete "in"

Response: Corrected (Line 341).

15. Line 345-368, this part is beyond scope of this paper because you were discussing "low pressure system". You might briefly mention the firework and CNY influence but not such the details.

Response: This part has been properly refined in manuscript at line 347-374 according to your comments.

[revised manuscript text omitted]